# Machine learning differentiates enzymatic and non-enzymatic metals in proteins

Ryan Feehan [1,3], Meghan W. Franklin[1,3] & Joanna S. G. Slusky [1,2✉]

Metalloenzymes are 40% of all enzymes and can perform all seven classes of enzyme reactions. Because of the physicochemical similarities between the active sites of metalloenzymes and inactive metal binding sites, it is challenging to differentiate between them. Yet distinguishing these two classes is critical for the identification of both native and designed enzymes. Because of similarities between catalytic and non-catalytic metal binding sites, finding physicochemical features that distinguish these two types of metal sites can indicate aspects that are critical to enzyme function. In this work, we develop the largest structural dataset of enzymatic and non-enzymatic metalloprotein sites to date. We then use a decision-tree ensemble machine learning model to classify metals bound to proteins as enzymatic or non-enzymatic with 92.2% precision and 90.1% recall. Our model scores electrostatic and pocket lining features as more important than pocket volume, despite the fact that volume is the most quantitatively different feature between enzyme and non-enzymatic sites. Finally, we find our model has overall better performance in a side-to-side comparison against other methods that differentiate enzymatic from non-enzymatic sequences. We anticipate that our model's ability to correctly identify which metal sites are responsible for enzymatic activity could enable identification of new enzymatic mechanisms and de novo enzyme design.

[1] Center for Computational Biology, The University of Kansas, Lawrence, KS, USA. [2] Department of Molecular Biosciences, The University of Kansas, Lawrence, KS, USA. [3] These authors contributed equally: Ryan Feehan, Meghan W. Franklin. ✉email: slusky@ku.edu

Enzymes are biological catalysts. They are known to increase reaction rates up to one million fold and facilitate reactions at biological conditions that would otherwise require high temperature and pressure. As of January 2021, 269,905 enzyme sequences[1] have been identified with 6,544 different reactions[2] and 108,391 protein structures[3]. These enzymes are classified by the Enzyme Commission (EC) with EC numbers categorizing each reaction[4]. Despite the importance and prevalence of enzymatic research, the ability to distinguish enzyme from non-enzyme sites remains an unsolved, challenging task.

Multiple physicochemical properties have been shown to be important predictors of catalytic function. Deviations in theoretical titration curves are able to identify active site residues responsible for Brønsted acid–base chemistry[5,6]. Bioinformatic studies have revealed that catalytic residues often lie in the largest surface-accessible cleft[7] that is closest to the protein centroid[8]. More recent work has shown that enzymes tend to have a network of amino acids that serve as a pathway for energy transfer[9,10]. Though these descriptive properties are helpful in enzyme site identification, they have low predictive value.

An appealing strategy for predicting if a protein or site is enzymatic is to use machine learning. Machine learning generalizes important trends from training data, which can be used to make future predictions on proteins with few or no homologs. Machine learning-based methods have won the last two critical assessments of methods of protein structure predictions[11–13], a competition for de novo structure prediction, demonstrating machine learning effectiveness of predicting protein characteristics on previously unstudied proteins. To date, there are two main types of machine learning algorithms related to enzyme prediction, enzyme function predictors which mostly use sequence data, and catalytic residue predictors which mostly use structure–data.

Machine learning methods that predict enzymatic function do so by producing EC numbers[14–20]. Due to the variety of enzyme functions, EC number predictors benefit from using multiple machine models, one of which predicts if a sequence is enzymatic or non-enzymatic, the next predicts the enzyme class, the next the enzyme subclass, etc.[15,17]. However, by taking a sequence level approach these algorithms miss critical active site information. A recent study demonstrated that machine learning EC number predictors and homology-based tools were rarely capable of distinguishing between native sequences and sequences where residues closest to the active site, eleven residues on average, were mutated to alanine[16].

An alternative to enzyme function prediction is enzyme active site prediction. Unlike the layered models required for EC number prediction, methods attempting to label catalytic residues output residue-based, binary, enzymatic, or non-enzymatic predictions[21–24]. When identifying residues responsible for a particular class of enzymatic function, structure-based features describing the 3D neighborhood have shown success[25,26]. Methods attempting to identify catalytic residues more generally, regardless of enzymatic function, benefit from combining sequence-based features that encapsulate the large amount of available data, such as conservation information, with structural-based features that describe the local environment[27]. These methods train on datasets using all enzyme residues, labeling only a few residues responsible for catalytic activity as positives. Such imbalanced datasets, where the positive samples are the minority class, result in low precision. In addition, comparing active site residues to protein surface and protein core residues may not predict catalytic activity so much as the existence of a pocket, as there are large differences in residue local environment unrelated to catalytic activity.

To address the challenges of differences in local environments, we focused on the metal ions of metalloproteins. Metals—whether enzymatic or not—often lie in pockets with unusual electrostatic properties[28,29]. The residues required for coordinating metals[30] are the same charged and polar amino acids commonly used for catalytic activity[31]. There are currently 53,987 crystal structures that are annotated to be metal-binding by metalPDB[32]. Considering that approximately 40% of enzymes are metalloenzymes[33], the overlap with the 108,391 structures with enzyme annotations leads to the creation of enzymatic and non-enzymatic target classes with a relatively low level of imbalance compared to residue-based datasets of enzymatic and non-enzymatic residues.

In this work, we create a homology-based pipeline that identifies metalloproteins and labels the metal-binding sites as enzymatic or non-enzymatic. The pipeline also removes proteins where the sequence is similar or metal-binding site is similar to prevent bias and overfitting during machine learning. We then calculate structure-based features describing the environment of the metal-binding sites, focusing on important catalytic properties. We use an agnostic machine learning strategy, training and optimizing several machine learning algorithms with different feature sets. The best model was determined by cross-validation and was evaluated on a holdout test-set, on which it achieved a 92.2% precision and 90.1% recall and identified enzymatic sites that were incorrectly labeled by our homology-based pipeline. We also examine the importance of the features used by our top model. Finally, we compare the performance of similar tools on our test-set, and find that our top model, using only structure-based physicochemical features, is overall superior to both enzyme function predictors and catalytic residue predictors.

## Results

**Data characteristics.** Metalloprotein crystal structures were queried from research collaboratory for structural bioinformatics (RCSB) and filtered for quality (Fig. 1a, details in methods section). Sites were defined as the area around a metal atom. Metal atoms in the same crystal structure file within 5 Å of each other were grouped as the same site. Sites were divided into two sets. Sites from structures deposited prior to 2018 were placed in the dataset used for algorithm optimization, model selection, and training the final model. Sites from structures deposited in 2018 or later were separated to form a holdout test-set, T-metal-site, which was only used to evaluate the final model.

Each site was labeled as enzymatic or non-enzymatic via a computational pipeline (Fig. 1b, details in methods section). Briefly, we identified metalloproteins that were homologous to enzymes in the manually curated mechanism and catalytic site atlas (M-CSA) database[34]. For further enzymatic evidence, we required that homologs met one of two conditions: associated EC number or an "ase" suffix in structure classification or molecular name. Finally, homologs were aligned with their respective M-CSA entries to check for structural homology and to label any sites adjacent to catalytic residues as enzymatic. Any remaining sites on homologs with enzymatic sites were labeled non-enzymatic. By finding homologs to the M-CSA proteins and then removing those with structural similarity we were able to identify 12,691 metal ions located at catalytic sites, 1089 of which were non-redundant, a nearly threefold increase from the 316 metal–ligand containing M-CSA entries (as of August 2019). Sites that were part of proteins that were not homologous to entries in the M-CSA and lacked all of the previously mentioned enzymatic evidence were classified as non-enzymatic. Because of the importance of correctly identifying enzymatic and non-enzymatic sites, any site that had some but not all of the enzymatic characteristics within our pipeline were discarded.

Biologically redundant data has shown to negatively impact machine learning models[35]. Having similar data in both the

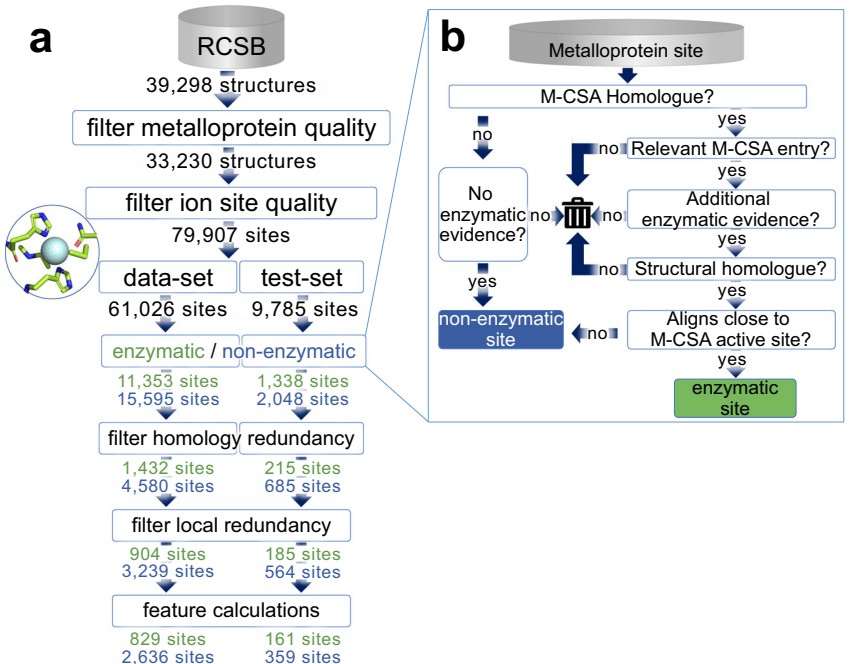

**Fig. 1 Workflow for dataset generation. a** The numbers on each arrow represent the number of entries present at that step; numbers of enzymatic sites are in green, non-enzymatic sites are in blue, and not enzymatically labeled entries are in black. The final numbers are representative of the end of the computational pipeline. **b** Computational labeling of sites as enzymatic/non-enzymatic via homology expansion of M-CSA (mechanism and catalytic site atlas) (see methods for more detail).

training and testing set can also lead to inflated performance evaluations. To prevent these issues, similarity within and between the test-set and dataset was reduced by filtering out sites with sequence or structural similarity. Briefly, within each set, sites were grouped according to homology. Within homologous groups, the metal-binding sites are compared by the similarity of coordinating residues. Only sites with unique residues were selected to represent each group. In this manner, homologs that have only mutated at the active site to perform different reactions will not be removed, but homologs with the same, conserved active site will be removed. Then in order to prevent similar proteins from being in both the dataset and test-set, we exhaustively compared all sites regardless of which set they were in, removing identical proteins and sites that were similar within a 6 Å sphere of the coordinated metal.

Our dataset used for ML is composed of 3465 sites from 2626 different protein structures; 24% of the sites are enzymatic (Supplementary Data 2). To check the accuracy of our pipeline labeling, we manually examined 50 sites labeled enzymatic and 50 sites labeled non-enzymatic. We found that all 50 sites labeled enzymatic were indeed catalytic active sites. In addition, three of the sites labeled non-enzymatic were in fact catalytic sites, giving our pipeline an estimated 97% balanced accuracy (Supplementary Data 1a). The test-set, T-metal-site, which is mutually exclusive from the dataset is composed of 520 sites from 404 different protein structures; 31% of the sites are enzymatic (Supplementary Data 2). Both sets contain sites distributed among the six major EC classes (Supplementary Fig. 1) excluding the translocases a class added to the EC after the start of this project.

Because our test-set is differentiated from our dataset by date of structure deposition, we assessed the two sets for covariate shift. Although advancements in crystallographic capabilities and changes in research funding that can affect the propensities of proteins deposited in the protein data bank (PDB) over time, no covariate shift was detectable between our final dataset and temporal test-set (Supplementary Methods).

**Feature analysis**. In an attempt to create a machine learning model that differentiates based on physicochemical information, we used physicochemical features, including those previously mentioned to describe the catalytic activity. However, we do not use features such as amino acid composition, conservation information, and secondary structure assignment[21,22] in order to avoid biasing any algorithm towards a specific fold. Moreover, metalloenzymes can be highly promiscuous[36,37], and assigning the correct metal can be tricky in a crystallized structure[38]; therefore, we also avoid using the bound metal identity.

In order to pass relevant, catalytic activity information to the machine learning algorithms, we developed a feature set with features from five categories: (1) Rosetta energy terms, (2) pocket geometry information, (3) terms describing the residues that line the pocket, (4) electrostatic terms, and (5) coordination geometry terms (Fig. 2a). Because Rosetta energy terms are residue-based and our features are site-based, different versions of the Rosetta energy term category were made by using two methods of aggregating each energy term—average or sum—and two methods of spatially defining the space around the sites—shells (0–3.5 Å, 3.5–5 Å, 5–7.5 Å, and 7.5–9 Å) or spheres (0–3.5 Å, 0–5 Å, 0–7.5 Å, and 0–9 Å) (Supplementary Fig. 2). In total, we used 391 features from the five categories (Supplementary Data 1b), though the features were not all used simultaneously during machine learning. To efficiently search the feature space, 67 combinations of feature sets ranging in size from 4 to 181 features were evaluated during model selection (Supplementary Data 1c).

To quantify the differences in feature values for enzyme and non-enzyme sites, we calculated the similarity between the enzymatic and non-enzymatic site values using a Jaccard index for discrete features and the percentage area of overlap between the kernel density estimates of the two feature curves for continuous features (Supplementary Fig. 3, details in methods section). Both methods lead to a scale where one is entirely the same between an enzymatic and non-enzymatic feature-value distribution and zero is entirely different between enzyme and

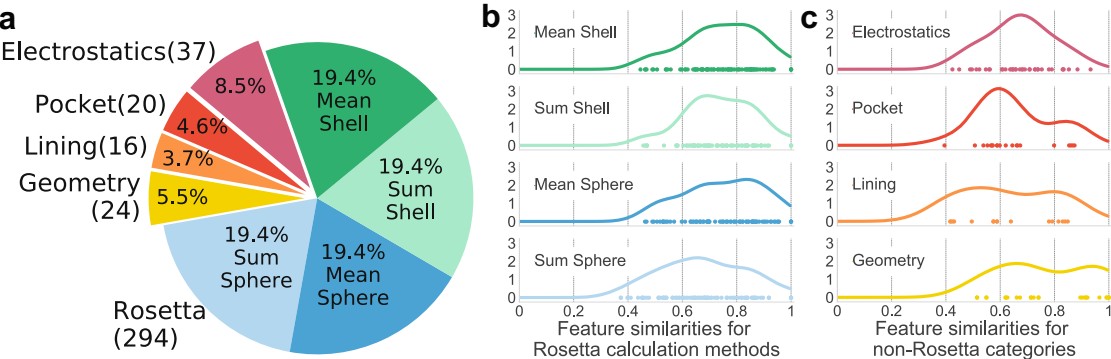

**Fig. 2 Relative size of feature categories and feature similarity distributions. a** Distribution of features used for training. The four groups of Rosetta terms each include 84 features calculated in one of four ways—the mean or average of residues within four shells or spheres—for a total of 294 unique Rosetta category features since the first shell and sphere are the same. **b**, **c** The kernel density estimations of feature similarity between enzymatic and non-enzymatic sites for each Rosetta calculation method (**b**) and the other four feature categories (**c**).

non-enzyme feature-value distribution. The Rosetta energy terms category that used sum sphere calculations had the most dissimilar features as demonstrated by having more points farther to the left (Fig. 2b fourth row). The sum calculation magnifies differences in features by not considering effects on a per residue basis and the sphere calculation considers more residues. Because catalytic sites tend to be closer to the protein centroid, more residues overall contribute to van der Waals and solvation terms and sum sphere calculations are especially dissimilar. In the electrostatics category (Fig. 2c first row), we find that the features farthest to the left are those that describe the deviations in the shape of theoretical titration curves for ionizable residues beyond the first shell. In the pocket category (Fig. 2c second row), the volume term is substantially more dissimilar than all other terms in that category. In the lining category (Fig. 2c third row), the most dissimilar features are those describing the number and volume of residues in the pocket lining. No geometry features (Fig. 2c fourth row) show particularly high dissimilarity between enzymatic and non-enzymatic sites.

**Machine-learning model optimization and selection.** To learn and then predict enzymatic and non-enzymatic sites, we selected fourteen classification algorithms from python scikit learn that span a variety of popular machine learning methods—support vector machines (SVMs), decision-tree ensemble methods, naive Bayes, nearest neighbors, a neural network, and linear and quadratic discriminant analysis (see Supplementary Methods for brief explanations).

Various scoring metrics are used to evaluate binary classification models. Imbalanced data (more non-enzymatic sites than enzymatic sites) can skew some of these metrics. For example, our dataset is 76% non-enzymatic sites. Therefore, we could achieve an accuracy of 76% by predicting non-enzymatic for all of our dataset. In order to prevent the imbalance of our dataset from biasing our evaluation metrics, we used the Matthews correlation coefficient (MCC, equation SE1)[39] that is less biased towards the imbalanced set. MCC values are on the [−1,1] interval, where any model predicting exclusively enzymatic or exclusively non-enzymatic would have an MCC of zero. The imbalance in our data could also produce an inflated recall, or percent of catalytic sites correctly identified, by over-predicting enzymatic sites without much of an effect on accuracy and true negative rate, or percent of non-enzymatic sites correctly identified. To avoid such over-predicting, we also prioritized precision, which is the percent of catalytic site predictions that are correct.

When a model learns the details of a training set too well, it can negatively impact the model performance on new data. This is called overfitting. Cross-validation (CV) is a common strategy that splits data into several different groups. Iteratively, one group at a time is left out for evaluation while the rest of the data is used for training. We used two CV techniques in a nested fashion (Supplementary Fig. 4), which allowed us to use different data for model optimization and model selection. Only the dataset was used for model optimization and model selection, allowing the test-set to act as a final model evaluation that was not be influenced by any previous training. During the inner CV, we optimized each algorithm for a specific feature set. We tested four different scoring metrics for optimization: accuracy, precision, MCC, or a multi-score combination of accuracy, MCC, and Jaccard index. In total, 3752 models were created (14 algorithms × 67 feature subsets × 4 optimization scoring metrics). We used the results from the outer CV to select the best of these models. However, 3274 of the models used different "optimal" versions of the machine learning algorithm during the outer CV. To eliminate any inflated metrics that may have come from this, we re-ran the outer CV using only the most frequently selected version of the algorithm for each model and discarded all models where large deviations persisted (Supplementary Fig. 5, details in methods section).

We graphed our remaining 1668 model performances by algorithm type (Fig. 3), optimization metric (Supplementary Fig. 6), Rosetta feature calculation method (Supplementary Fig. 7), feature category exclusion, and feature set size (Supplementary Fig. 8). The only emerging trends were based on machine learning algorithm type. The neural network (neural network, Fig. 3 tan) and decision-tree ensemble methods (extra trees, gradient boosting, and random forest, Fig. 3 blues) perform most favorably for our prioritized metrics, MCC, and precision. SVMs and linear models (logistic regression, ridge, and passive-aggressive Fig. 3 purples) had the highest recall (Supplementary Fig. 9). However, the relatively low precision of the SVM and linear models indicates that these high recall values are the result of over-predicting enzymatic sites.

**Top model evaluation.** The top model in our model selection was an extra-trees algorithm using all feature categories with mean, sphere calculations for Rosetta terms. We named this model Metal Activity Heuristic of Metalloprotein and Enzymatic Sites (MAHOMES) (Fig. 3 "X"). In addition to having the best precision relative to MCC, MAHOMES was also surprisingly stable. Three out of the four optimization strategies selected the same hyperparameter set on all seven CV folds, indicating that MAHOMES performance is not an overfit optimistic evaluation.

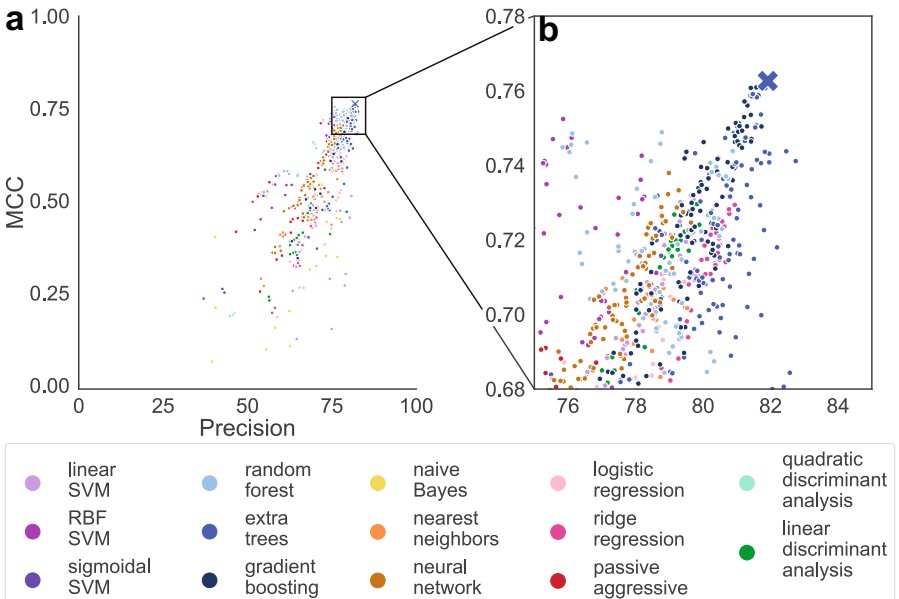

**Fig. 3 Outer CV performance by the algorithm. a** Each point represents the results for a specific model. Points are colored according to the algorithm used and grouped by classifier type; support vector machines (SVMs) are purples, decision-tree ensemble methods are blues, linear models are reds, discriminant analysis is greens, no grouping for naive Bayes, nearest neighbor, and neural network. Better performing classifiers should be close to the upper right corner. The X denotes our top model (extra trees with AllMeanSph feature set). **b** Zoomed-in view of boxed region in (**a**).

A final evaluation of MAHOMES was performed using the T-metal-site test-set (2018–2020 structures) where it achieved slightly higher performance metrics than its outer CV performance (Supplementary Table 1). The final performance evaluation still falls within projected deviation, as observed on different test folds during outer CV, supporting the validity of the reported performance metrics.

We manually inspected the sites misclassified by MAHOMES. This included 27 of the 359 non-enzymatic sites misclassified as enzymatic (false positives) and 17 of the 161 enzymatic sites misclassified as non-enzymatic sites (false negatives) (Supplementary Data 1d). Manual inspection of these sites in the literature revealed that ten of the 27 sites that had been labeled by the pipeline as non-enzymatic but as enzymatic by MAHOMES were actually correctly predicted by MAHOMES and incorrectly identified by the pipeline (Supplementary Data 1d green). All ten cases in which MAHOMES correctly predicted sites misclassified by the pipeline were sites that lacked M-CSA homologs EC numbers, and an "ase" suffix. Eight of the sites were for proteins that had not been structurally resolved before and had no homologs in M-CSA. The two sites bound to proteins that had previously been solved were both bound to sonic hedgehog proteins. Though the zinc domain of sonic hedgehog proteins was previously thought to be a pseudo-active site, more recent research indicates that it is responsible for metalloprotease activity[40,41]. In addition to the ten mislabeled sites, we were unable to definitively determine the correct label for four of 27 the false-positive sites because they are not well characterized by the literature (Supplementary Data 1d yellow). We recalculated the T-metal-site performance metrics by changing the ten pipeline mislabeled sites to be true positives and excluding the four false positives we could not definitely label (Table 1).

**Feature importance**. Since MAHOMES is a decision-tree ensemble algorithm, it is capable of producing relative feature importance for classification (Supplementary Data 1e). By graphing feature importance against our previously calculated similarity, we find that MAHOMES did not find features with high similarity to be useful in differentiating enzymatic and non-enzymatic sites (Fig. 4). However, lower similarity did not always translate to higher feature importance. For example, the lowest similarity in this feature set was volume, with a similarity of 39%, meaning it is quantitatively the most useful feature, supporting previous reports of its utility for describing enzymatic active sites[7]. However, volume was only the 13th most important feature for enzyme classification. So, MAHOMES finds twelve other features to be more valuable for differentiating enzymatic activity despite having more similar values for enzymatic and non-enzymatic sites. For example, the feature of the distance between the center of the active site and protein centroid[8] is the ninth most important feature for discrimination despite a 57% feature similarity.

The most important feature for MAHOMES is an electrostatic feature derived from the shape of the theoretical titration curve of local ionizable amino acids. Specifically, it is the average of the second moment of the theoretical titration curve derivative for second shell residues (3.5–9 Å) from the metal site. This feature was implemented due to the previous findings that residues at the active site have a higher deviation from the Henderson–Hasselbalch equation than other ionizable residues[5,6,23,42,43]. The features of the averages of the third and fourth moments for second shell ionizable residues, while still important, were less critical, ranking 10th and 19th, respectively. The electrostatics of the residues responsible for coordinating the metal ions (first shell) are more similar between the metalloenzyme and non-enzymatic metalloprotein sites (Supplementary Data 1b) than in the second shell and this is likely preventing these descriptors from being as important. The other four of the top five features are pocket lining terms describing the number and total volume of amino acids lining the pocket.

Although MAHOMES found the previously mentioned features to be the most helpful for predicting enzymatic activity, they are not solely responsible for enzymatic activity. During the outer CV, which evaluated 3465 predictions, an extra trees model excluding the electrostatic category only made six more incorrect predictions

**Table 1 Comparison of MAHOMES performance to similar tools that make enzymatic and non-enzymatic predictions.**

| Method | Predictions by | Evaluation set | Accuracy (%) | Precision (%) | Recall (%) |
|---|---|---|---|---|---|
| MAHOMES (this paper)[a] | Metal-binding site | Corrected T-metal-site | 94.2 | 92.2 | 90.1 |
| DeepEC[a,16] | Sequence | T-metal-seq | 69.9 | 59.6 | 90.5 |
| DEEPre[a,44] | Sequence | T-metal-seq | 90.1 | 81.3 | 100.0 |
| EFICAz2.5[a,45] | Sequence | T-metal-seq | 90.8 | 88.4 | 90.0 |
| PREvaIL[b,21] | Residue | T-124 set | 96.8 | 14.9 | 62.2 |
| CRHunter[b,22] | Residue | T-124 set | 98.6 | 28.6 | 48.8 |
| CRPred[b,24] | Residue | T-124 set | 97.3 | 14.7 | 50.1 |

True positive, true negative, false positives, and false negatives for the MAHOMES and the three enzyme function predictors in Supplementary Table 2.
[a]The enzyme function predictors evaluated on a comparable set to MAHOMES.
[b]The enzyme site predictors previously tested on the independent, holdout T-124 set[24].

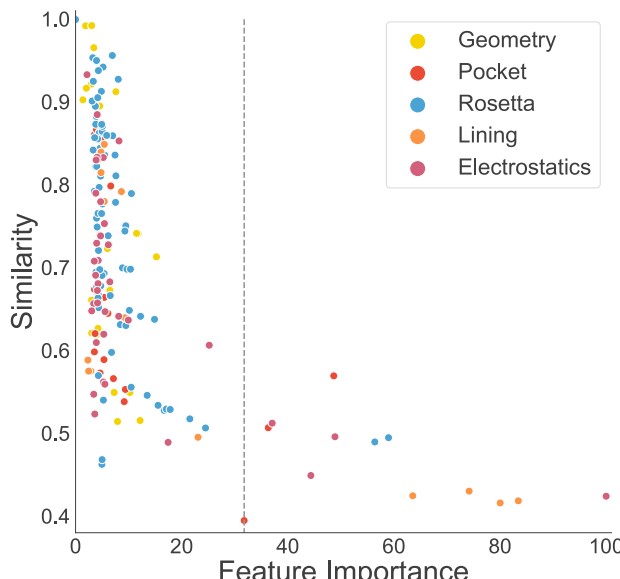

**Fig. 4 Feature similarity with respect to feature importance.** Each point represents a feature and is colored according to the feature category: geometry features are yellow, pocket features are red, Rosetta features are blue, lining features are orange, and electrostatic features are purple. The y-axis is the similarity calculation for the enzymatic and non-enzymatic feature values (see methods for more detail). The x-axis is the features importance for MAHOMES (see methods for more detail). The dashed line indicates the feature importance of volume. See Supplementary Data 1e for values and feature names.

than MAHOMES (three enzymatic and three non-enzymatic). Another extra trees model excluding the pocket lining category correctly identified two more non-enzymatic sites than MAHOMES, but it also identified fourteen fewer enzymatic sites.

**Benchmarking with other methods.** Since no alternative method uses metal-binding sites as input, we adjusted the corrected T-metal-site test-set to be a sequence test-set (T-metal-seq) to compare the performance of MAHOMES to other methods. The 516 metal sites in the corrected T-metal-site test-set are on only 400 proteins that were unambiguously metalloenzymes or non-enzymatic metalloproteins. The T-metal-seq test-set consisted of those 400 unique sequences (Supplementary Data 3). Sequences were labeled as enzymatic or non-enzymatic as described in the methods. Using T-metal-seq, we benchmarked the performance of MAHOMES against three enzyme function predictors (Table 1); DeepEC[16], DEEPre[17,44], and EFICAz2.5[45]. MCC is sensitive to the magnitude of imbalance, so it was not used as a

performance metric due to the different levels of imbalance in the corrected test-set and sequence test-set.

DeepEC[16] predicts enzymatic function using three independent convolutional neural networks which only use protein sequence as input. The first neural network makes a binary enzymatic or non-enzymatic prediction. We only evaluated the performance of the first neural network since the second and third neural networks make third and fourth-level EC number predictions. DeepEC had a similar recall to MAHOMES but was by far the lowest of the evaluated methods for accuracy, and precision.

Similar to DeepEC, DEEPre[17,44] also uses deep learning to predict enzymatic function. DEEPre is a level-by-level predictor, meaning it has a machine learning model for every split in the EC hierarchy for EC numbers that it is capable of predicting. We only evaluate its level zero model, which is responsible for making enzyme or non-enzyme predictions. In addition to the sequence, DEEPre uses some features mined from the sequence data, including predicted solvent accessibility, predicted secondary structure, and evolutionary information. The evolutionary information includes the detection of any Pfam functional domains and a position-specific scoring matrix for the whole sequence produced by BLAST+. DEEPre had a remarkable 100% recall, identifying all enzyme sequences including the ones mislabeled by our pipeline. However, it over-predicted the number of enzymatic sequences by 23%, resulting in a lower precision and accuracy than MAHOMES.

The final sequence-based enzyme function prediction tool evaluated on our sequence test-set was EFICAz2.5[45–47]. EFICAz2.5 combined homology detection, sequence similarity, conservation information, Pfam functional domains, the PROSITE database to generate four independent predictions and two predictions made by SVMs. These six outputs are combined using a tree-based classification model which outputs an EC number. EFICAz2.5 was very consistent across all evaluation metrics. Although MAHOMES had the highest precision and true negative rate, EFICAz2.5's came in a close second relative to DeepEC and DEEPre.

## Discussion

**Feature importance.** Many of the features of most importance to MAHOMES are similar to features previously described for determining active sites. Our most important electrostatic features are modeled on previous electrostatic features[5,6,23,42,43] but have subtle differences. Both use calculations of the moments for the theoretical titration curve first derivative. The previous work identifies active site residues by looking for clusters of residues with statistically significant deviations. We find the averages of deviations for ionizable residues in the area beyond the coordinating residues (3.5–9 Å) identifies metalloenzyme catalytic sites over sites that only bind metals. We find that the coordinating

residue theoretical titration curves are more similar between enzyme and non-enzyme, likely because both our enzymatic and non-enzymatic sites are coordinating metals. The differences beyond the first shell likely indicate a network of amino acids that serve as a pathway for energy transfer[48] which have been found for a number of types of enzymes[9]. In addition, we also find the second moment to be most predictive whereas other studies find the third and fourth moment to be most predictive[6]. Further investigation will be needed to determine the origin of this difference.

Our volume feature, modeled on a previous study[7], was the most dissimilar feature between enzymatic and non-enzymatic sites but was far from the most important feature for activity prediction. In contrast, a feature of the distance between the center of the active site and protein centroid, also modeled on the previous work[8], was 1.5 times as similar between classes as the volume feature but was a more important feature for activity prediction.

The second to fifth most important features are pocket lining features describing the number and volume of side chains and backbones lining the pocket. In all four of these features, the enzyme dataset has more volume and more amino acids. There are two possible explanations for the importance of this feature. The first is that it may indicate that the surface area of the pocket is superior to the volume of the pocket in predicting enzymatic activity. The second is that because of the movement and flexibility known to be critical to enzyme function[9] the amino acids lining the pocket are a better proxy for the true volume of the pocket than the volume calculation itself. Further research may help deconvolute these two possibilities to better explain what about the pocket surface is most predictive of enzyme activity.

**MAHOMES in comparison to catalytic residue predictors**. Though catalytic residue predictors use structure-based features and therefore may be more comparable to MAHOMES, we were unable to directly benchmark MAHOMES performance against any catalytic residue predictors[22,49,50,21,24] due to lack of availability of either the model or the methods for implementation. Models such as these which train on very imbalanced datasets (an enzymatic to the non-enzymatic ratio of ~1:124) result in misleadingly low precision and misleadingly high accuracy. Precision is the percent predicted enzymatic sites that were correct, when the number of correct enzyme sites is low compared to the total possibilities, this number will always be low. Conversely, accuracy, which is the percent of predictions the model correctly predicts, will be very high for an imbalanced set. For example, a 99.2% accuracy may appear to represent a successful predictor but could be achieved with only non-enzymatic predictions. A more equitable comparison to MAHOMES would be recall. CRPred[24], CRHunter[22], and PreVAIL[21] were all evaluated on the T-124[24] dataset. Because the common dataset contains a similar number of enzymatic sites as our test-set, recall—the percent enzymatic residues that were correctly identified—is a more equitable comparison. These models report recalls ranging from 48.8 to 62.2%, whereas MAHOMES scored 90.1% on recall. We did not compare another catalytic residue predictor, POOL[23,49] because they did not use the same independent, holdout test-set.

As measured by the difference in recall MAHOMES can correctly identify 1.3 times more catalytic sites on enzyme structures than the best of these three catalytic residue prediction methods. Moreover, where MAHOMES predicts nine out of every ten enzymatic predictions correct, catalytic residue predictors are only correct for one or two out of every six enzymatic predictions. We anticipate that MAHOMES relative success is due both to

training on more similar sites and to less imbalance of the training set. By training on negative sites that were also in pockets and also coordinated metals MAHOMES was able to assign feature importance based on characteristics that were particular to enzyme activity. In addition, MAHOMES training set was up to 40 times more balanced than the catalytic residue predictor datasets.

**MAHOMES in comparison to enzyme function predictors**. Side-to-side comparison with our test-set demonstrated that MAHOMES also had overall better performance at predicting enzymatic activity for metalloproteins than sequence-based enzyme function prediction tools. The one exception is the 100% recall by DEEPre which was better than MAHOMES 90.1% recall. This is a reflection of DEEPre over-predicting enzymatic sites, as indicated by its lower precision (81.3%) which is the percent of predicted enzymes that were correctly identified. The other neural network model, DeepEC, though less successful than DEEPre had a similar problem of over-predicting enzymatic sites. DeepEC scored a 90.5% recall but only a 59.6% precision. Conversely, MAHOMES had a more balanced performance with a 92.2% precision and 90.1% recall. EFICAz2.5, the most homology-based predictor, showed that not all EC number predictors suffer from enzymatic over-predictions with a 90.0% recall, 88.4% precision, but it was still outperformed by MAHOMES for all performance metrics.

Differences in what features each tool used to indicate what can be used to successfully make enzymatic and non-enzymatic predictions. DeepEC relies heavily on deep learning, passing only the protein's sequence to its predictor with no other processing or features. DeepEC's fourth-place performance indicates that relying on sequence alone requires even more training data to allow for deep learning to extrapolate important features from the sequence. The other two sequence-based methods generate evolutionary information from the sequences which is combined with machine learning. Their large training sets, 44,316 sequences for DEEPre and 220,485 sequences for EFICAz2.5, provided ample data to allow for successful enzymatic and non-enzymatic predictions via homology and conservation.

MAHOMES structure-based, physicochemical features led to predictions by features specific to catalytic activity, allowing MAHOMES to outperform DEEPre and EFICAz2.5 even though it was trained with only 7.8% and 1.6% the amount of data, respectively. We anticipate that MAHOMES success over enzyme function classifiers was due to the use of structural features which are more sensitive to small differences of the active site. Though structural data is less available, it is more predictive of enzyme function than homology.

MAHOMES ability to correctly detect enzymatic activity should make it especially useful to problems where methods reliant on homology are not applicable. For example, MAHOMES may be used for eliminating poor de novo metalloenzyme designs prior to experimental testing. Another use-case could be detecting when a mutation eliminates enzymatic activity. As more datasets become available, we anticipate that the MAHOMES approach will be able to be refined and deployed in future protein design and catalytic-site prediction pipelines.

## Methods

**Metalloprotein ion site data**. For the purposes of our study, we focused on protein structures that contain residues codes with one or two of the following metals: iron, copper, zinc, manganese, magnesium, molybdenum, nickel, and cobalt (residue codes of interest are FE, FE2, FES, FEO, CU, CU1, CUA, MG, ZN, MN, MO, MOO, MOS, NI, 3CO, and CO).

RCSB[3] was queried to find the list of crystal structures containing a residue code of interest. Structures with nucleotide macromolecular types were removed. Crystal structures with a resolution greater than 3.5 Å were removed to ensure high-quality

feature calculations. Structures with more than 40 sites were removed as those indicated large complex structures such as virus particles and ribosomes. Protein chains with less than 20 residues were removed to avoid polypeptides. For cross-referencing and to eliminate as many de novo proteins or immunoglobulins as possible, we also removed protein chains that did not have a UniProtKB accession number[1].

For uniform handling of mononuclear and polynuclear metal sites, we defined a site as the area around metal atoms of interest attached to the same protein chain within 5 Å of each other. Because homology detection was determined by a chain, sites that were within 4 Å of a residue from another chain were indicative of a possible multi-chain metal-binding site and were removed. In addition, the few sites with more than four metal atoms, such as metal storage proteins, were removed to allow for a more uniform feature calculation methodology. Sites within 10 Å of mutated residues were removed to avoid studies that identified catalytic residues through loss of activity mutations. We split the sites into a dataset and a test-set—the dataset consisted of protein structures resolved prior to 2018; the holdout test-set, T-metal-site, consisted of protein structures resolved in 2018 or later.

**Computational labeling of sites as enzymatic and non-enzymatic.** The M-CSA contains manually annotated reaction mechanisms with a representative protein structure in the PDB for each entry[34]. A few M-CSA entries are undesirable for our studies, such as those requiring heme cofactors or dimerization. Additionally, a few M-CSA entries' catalytic residues were missing metal coordinating residues or included unwanted residues, such as those with steric roles. We performed our own manual annotation to adjust the data we used in these cases (Supplementary Data 1f). To expand the coverage of the M-CSA data, potential sequence homologs were detected for each M-CSA entry using PHMMER[51] with a cutoff E-value of $10^{-6}$ and a database of sequences in the PDB. E-values take into account database prevalence, leading to the addition and removal of detected homologs when using updated versions of the PDB. Hence, two versions of the PDB are used to allow for updating homolog detection for the test-set (PDB as of May 21, 2020) without affecting the homologs detected in the dataset (PDB as of May 17, 2019). Homologs of the undesirable M-CSA entries were removed from the dataset and test-set. It is estimated that between 0.5% and 4% of M-CSA homologs are likely to be pseudoenzymes or non-catalytic enzyme homologs[52]. To avoid labeling pseudoenzymes, we discarded homologs that did not meet at least one of the following requirements; an associated EC number, a PDB structure file classification containing "ase", or a PDB structure file macromolecule name containing "ase".

For further support of enzymatic evidence and identification of the homolog's active site location, each remaining homolog was aligned to its respective M-CSA representative, using TM-align[53]. We chose to use a TM-score of 0.40 or greater to represent structural homology and discarded all aligned homologs below that cutoff. All sites that met all previous requirements and aligned within 5 Å of a catalytic residue listed in the M-CSA entry were labeled as enzymatic. For chains containing enzymatic sites, unlabeled sites were labeled as non-enzymatic. To create our non-enzymatic set, sites that were not M-CSA sequence homologs, had no associated EC number, no "ase" in the PDB structure file classification, and no "ase" in the PDB structure file macromolecule name were labeled non-enzymatic. The remaining unlabeled sites in the dataset and test-set were removed at this point.

The non-enzymatic metal-binding sites comprise a number of types of proteins some of which may be better suited for the negative dataset than others. The most frequent types of proteins are classified as transcription, transport, signaling, and binding (proteins, DNA, RNA, sugar, or membrane). Metal transport and signaling proteins are likely to only have pockets large enough to fit the metal ions. In addition, the non-enzymatic metal ions that do not contribute to the protein function are likely to either be of structural importance or crystal artifacts. The metals bound as crystal artifacts are likely to be bound at the surface and bound less tightly than metalloenzymes.

**Removal of redundant sites.** There are many redundant protein structures— proteins that are highly homologous or identical proteins with different ligands. To prevent redundancy from biasing our training toward one particular protein we implemented the following method of intra-set redundancy removal. First, we removed proteins that were sequence redundant. We removed all but one instance of identical sequences. Then sequence homologs were assembled and clustered using PHMMER taking from the PDB on the dates mentioned and using an E-value of $10^{-20}$. This homolog collection and clustering were executed independently for the dataset and test-set. Next, we removed site redundancy. Chains within a cluster were aligned using TMalign and sites that aligned within 5 Å with a ≥0.50 TMscore were checked for local similarity. We defined local site similarity as the Jaccard index of residue identities within a 3.75 Å radius for two sites. The Jaccard index is a similarity metric defined by Eq. (1), where **A** and **B** are the vector of amino acid identities surrounding two different sites.

$$J(\mathbf{A}, \mathbf{B}) = \frac{|\mathbf{A} \cap \mathbf{B}|}{|\mathbf{A} \cup \mathbf{B}|} \qquad (1)$$

This results in a value of 0 when there is no similarity and a value of 1 for identical sites. Sites were removed if they had a local similarity greater than 0.80. Due to high preprocessing computational costs, sites that had already undergone

relaxation were selected when possible. Otherwise, we used the following priority to keep the more favorable site: (1) catalytic site, (2) no other ligands present, (3) no other sites within 5–15 Å, (4) no mutations in protein structure, and (5) crystal structure resolution.

Evaluating machine-learning models on the same data used to train and optimize them can lead to overfitting and inflated metrics. The temporal separation of the test-set and dataset prevents the same structure from being in both sets. However, the aforementioned redundancy of protein structures, different structures of the same protein, or closely related homologs can still lead to reporting an inaccurately high machine learning performance. To remove structurally similar sites, we used an all against all method to compare the residue identities within a 6.0 Å radius of all remaining sites, removing sites with greater than 0.80 Jaccard similarities.

**Further data processing.** The pipeline described above displayed catalytic identification errors with NTPases, in part because they were inconstantly labeled in the training dataset. To remove all ATPase and GTPase sites, we removed all Mg sites within 10 Å of a nucleic acid-phosphate ligand that were labeled non-enzymatic. Finally, sites in which the metal is not well coordinated are likely due to crystal artifacts and are poor negative controls for metalloenzyme sites. When the structures were relaxed using Rosetta (see Supplementary Methods for RosettaScripts inputs), 728 sites with loosely bound metals—often the result of crystal artifacts—that moved more than 3 Å during relaxation were removed from the dataset and test-set. Also, 179 sites were removed due to failure to complete feature calculations. The remaining metals had a similar distribution of a number of coordinating atoms between enzymatic and non-enzymatic sites (Supplementary Fig. 10).

**Features.** Five feature categories were calculated on the relaxed structures— Rosetta energy terms, pocket void, pocket lining, electrostatics, and coordination geometry (Fig. 2A; Supplementary Data 1b). To prevent outlier values from affecting the models, all features were scaled (normalized) using sci-kit's RobustScaler prior to machine learning. The scaler was fit to the 20th and 80th quantile of the dataset and used to transform the dataset and test-set.

**Rosetta energy terms.** Rosetta feature values were assigned to all sites using all the terms in the energy function *beta_nov16*[54]. Rosetta assigns a value for each term to each residue. We used the sum or the mean of the per-residue Rosetta energy terms as features, calculating terms in spheres and in shells using the cutoffs 3.5, 5, 7.5, and 9 Å. This results in 21 features per cutoff and 84 features per calculation method. The different groups of Rosetta terms were never included together in any model.

**Pocket void terms.** Rosetta's pocket_measure application[55] was executed on all individual sites, using the residue closest to the site center of the anchor the grid. This maps the pocket to a 0.5 Å-interval grid. Using these pocket grid points, we determined the city block distance from site center to the center of the pocket opening. Volumes were calculated by Rosetta and depth was taken as the distance between pocket opening and site center.

To quantify the shape of the pocket, we took three slices of pocket points (bottom, middle, and top) and projected them into 2D space after rotating the pocket so that the z-axis runs from site center to center of the pocket opening. For each slice, we calculated the farthest distance between two points, the area formed by a 2D slice that encompassed all points, the distance from the origin to the center of the ellipse that encompasses the convex hull, and the two radii of the ellipse. These calculations result in 20 features.

**Pocket lining terms.** The grid produced by Rosetta's pocket_measure also allows us to identify and describe the residues that line the pocket. Residues were split into backbone-only—where exclusively backbone atoms are adjacent (within 2.2 Å) to a pocket grid point—and sidechain—where any sidechain atom of a residue is adjacent to a pocket grid point; adjacent sidechain and backbone atoms of the same residue are included in this group). We then calculated the average, minimum, maximum, skew, and standard deviation of the hydrophobicity for the side chain residues using two different scales—Eisenberg[56] and Kyte–Doolittle[57]. We also calculated the occluding volume of the pocket as the sum of the van der Waals volume of the sidechains in the pocket which allowed us to determine the total volume of the pocket without sidechains and the Leonard Jones volume of that pocket occupied by sidechains. Finally, we calculated the surface area of the pocket walls by summing the solvent-accessible surface area for any sidechain-adjacent residue. This resulted in 16 features.

**Electrostatics.** Our electrostatics features are based on the use of theoretical titration curves by THEMATICS[5,23] which showed that the theoretical pKa curves of ionizable side chains deviate from expected Henderson-Hasselbach behavior[42]. We used bluues[58] to calculate our theoretical titration curves for all ionizable residues (see Supplementary Methods for command-line options), which is a generalized Born solution to the Poisson-Boltzmann equation rather than a finite

difference solution and therefore quickly calculates the electrostatics for our dataset.

We calculated the mean and max of the second, third, and fourth central moments of the theoretical titration curves for all residues in two shells. The first shell included residues with an α-carbon atom within 3.5 Å of the site center; the second shell included residues with an α-carbon atom between 3.5 and 9 Å of the site center; the shells are labeled "Inside" and "Outside", respectively. Each scaler of feature, the second, third, and fourth central moments, was also used to calculate an environmental feature[43]. For a scaler feature $x$ and a site center $\mathbf{s}$, the corresponding environmental feature was calculated using Eq. (2).

$$x^{\mathrm{env}}(\mathbf{s}) = \frac{\sum_r w(\mathbf{r}) x(\mathbf{r})}{\sum_r w(\mathbf{r})} \qquad (2)$$

where $\mathbf{r}$ is an ionizable residue with a distance $d(\mathbf{r},\mathbf{s}) < 9$ Å from the site center, and the weight $w(\mathbf{r})$ is the $1/d(\mathbf{r},\mathbf{s})^2$.

Bluues[58] also provides information about the pKa shift from an ideal amino acid to the amino acid pKa in the context of neighboring ionizable residues. Because pKa shifts are observed in catalytic residues[59–61], we include the minimum and maximum of these values for the same two shells as the central moments described above. We also calculated the residuals for the deviation of the theoretical titration curve from a sigmoidal curve; we similarly calculated the mean and max of these values in the two shells as described above. Residues adjacent to active sites often rank among the most destabilizing (positive) $\Delta G_{\mathrm{elec}}$ values of a protein[62]; we use the solvation energies calculated by blues as a proxy and rank all residues from highest to lowest solvation energies. Residue ranks are then split into five bins to avoid length-dependent ranks; destabilizing ranks run from highest to lowest solvation energies while stabilizing ranks were assigned from lowest to highest solvation energies. We then calculated the mean and max in the two shells as well as the environmental rank as described above. Overall, there are 37 electrostatic features.

**Coordination geometry.** FindGeo[63] and CheckMyMetal[64,65] are both webservers that provide information about the coordination geometry of bound metals in crystal structures. We added functionality to FindGeo's python portion to calculate features from CheckMyMetal. FindGeo compares the coordination of a metal atom in a protein structure to a library of 36 idealized geometries and, in addition to the name of the coordination geometry, determines whether the geometry is irregular, regular, or distorted. For each site, we record which of the 36 geometries are present. However, these features are not used in training because assignments were diverse enough that it leads to problems of sparse data. Instead, we recorded coordination numbers based on the 36 ideal geometries and determined whether the geometry included a vacancy.

We also recorded the numbers of nitrogen, oxygen, sulfur, and other atoms coordinating the metals in each site. Both the total number of coordinating atoms and the average number per metal are included as features. We calculated the overall charge on the site from the FORMUL lines of the PDB structure file. Four terms from CheckMyMetal are also calculated—the sum of the angular deviation from the ideal geometry, the maximum single angle deviation, the bond valence (which is the sum of the individual valences), and the normalized bond valence. Perfect geometries are defined to have a normalized bond valence of 1. These features all attempt to describe how far from physically ideal the metal-binding site is. In total there are 24 coordination geometry features.

**Feature similarity.** We calculated feature similarity for discrete features—those that can take fewer than 21 unique values—using the Jaccard similarity between the proportions observed in the enzymatic and non-enzymatic sites. In this case, the Jaccard similarity is equal to the sum of the minimum values divided by the sum of the maximum values. For example, if 50% of the non-enzymatic sites have regular geometry and 50% do not, and 75% of the enzymatic sites have regular geometry and 25% do not, the Jaccard similarity is $(0.5 + 0.25)/(0.5 + 0.75) = 0.6$.

We calculated feature similarity for our continuous features, as the overlap coefficient which is the area under the minimum of two curves or the shared area of two curves. First, we fit a kernel density estimator (KDE) to the enzymatic values and to the non-enzymatic values. The KDEs were evaluated at $2^{10} + 1$ data points equally spaced from the minimum and maximum values of the feature across the whole training set. Then, Romberg integration was used to calculate the area under the minimum of the two evaluated KDEs.

**Machine learning.** We selected fourteen classification algorithms[66] readily available in Python[67] that cover a variety of popular methods for machine learning — linear regression, decision-tree ensemble methods, SVMs, nearest neighbors, Bayesian classification, and simple neural networks. A hyper-parameter search space of 8–12 hyperparameter sets was selected for each algorithm (algorithms and search space can be found at https://github.com/SluskyLab/MAHOMES). The algorithms require or greatly benefit from normalizing the feature values. We used the sci-kit RobustScaler with the 20th and 80th quantiles to limit the effect of outliers during scaling. The scaler was fit to the dataset and used to transform both the dataset and test-set. Due to the imbalance of target classes (more non-enzymes

than enzymes), the dataset was randomly under-sampled at a ratio of three non-enzymatic sites to one enzymatic site.

A nested cross-validation strategy was used for model optimization to avoid overfitting and bias (Supplementary Fig. 4). Each inner loop used GridSearch with StratifiedShuffleSplit (in the python scikit-learn package[67]) and was optimized four times for each of four scoring terms—accuracy, precision, MCC, or a multi-score combination of accuracy, MCC and Jaccard index. The outer loop CV used stratified k-fold cross-validation. The most frequently selected hyperparameter set during the outer cross-validation was considered optimal for the model. The dataset was under-sampled once prior to model optimization.

In total, we examined 3,752 machine learning models (14 algorithms × 67 feature sets × 4 optimization terms). For model selection, we re-ran the outer cross-validation using only the optimal hyper-parameter set. During stratified k-fold cross-validation, the data is divided into $k$ groups ($k = 7$), each with the same number of positive and negative entries. All except for one of the groups are used to train a model and the left out group is used to evaluate that model. This is repeated $k$ times, leaving out a different group each time, essentially testing the model on seven different subsets. The performance is then averaged. Our random sampling, and some of the machine learning algorithms that require random sampling, are susceptible to differences in the machines on which they are executed. In order to produce more reliable performance evaluations for model selection, we repeated each iteration of the outer cross-validation ten times when we re-ran it. During each repetition, a new random seed was passed to the machine learning algorithm and used to under-sample the training folds. Since we used $k = 7$, the reported outer cross-validation results are the average of 70 different models (7-folds, each with 10 different random seeds).

The second run of the outer cross-validation resulted in a much higher performance deviation for the different folds (Supplementary Fig. 5), supporting that a large number of models had overfitted evaluations due to changing hyperparameter sets during the initial outer cross-validation. To avoid selecting a model that only performed well for specific data, we filtered the results to keep models that met several conditions: Accuracy standard deviation $\leq 6.5\%$, True Negative Rate standard deviation $\leq 9\%$, and MCC standard deviation $\leq 0.11$. Only 1668 of 3752 models considered met these requirements (Supplementary Fig. 5). Our top model was an extra-trees algorithm using all feature categories with mean, sphere calculations for Rosetta terms because it had high MCC, high precision, and converged on the same optimal extra trees algorithm for three of the four optimization metrics.

A holdout test-set was used for our final performance evaluation of the selected top model. Similar to the second outer cross-validation run, we repeated the predictions ten times. Each repetition used a new random seed for the extra tree's algorithm and a different random under-sampling of the non-enzymatic dataset sites. If the average of the ten predictions was <0.5 the site was classified as non-enzymatic if the average was ≥0.5 the site was considered enzymatic.

Finally, we used the scikit-learn ExtraTreesClassifier feature importance to output the impurity-based feature importance. This feature importance is calculated based on how much each feature contributed to the reduction of the Gini criterion for the extra trees model. During each repetition, the output impurity-based feature importance was normalized relative to the largest value, giving feature importance between 0 and 100. The results from the ten repeated runs with different random sampling were averaged to give the final feature importance (Supplementary Data 1e).

**DeepEC, DEEPre, and ENZYMAz2.5 method evaluations.** After removing the inconclusive false positives from the test-set, the set contained 516 sites on 400 unique sequences. The sequences were used for testing sequence-based classifiers for metrics comparison. Because sequences could contain both positive and negative metal-binding sites, sequences were labeled as enzymatic if they met at least one of the following criteria: an associated EC number, an "ase" suffix in structure classification or molecular name, or previously identified to be enzymatic by manual inspection. Sequences that did not meet any of the previous criteria were labeled as non-enzymatic. We manually verified 33 enzymatic labels for sequences that lacked an associated EC number. The sequence test-set contained 170 enzymatic sequences and 230 non-enzymatic sequences.

EFICAz2.5 was download from https://sites.gatech.edu/cssb/eficaz-5/. We used it to make a prediction for all 400 sequences in the sequence test-set. Predictions containing "No EFICAz EC assignment" were considered non-enzymatic and all other predictions were considered enzymatic. DeepEC was downloaded from https://bitbucket.org/kaistsystemsbiology/deepec/src/master/. DeepEC was only able to make predictions for the 396 sequences that were less than 1000 residues in length. The output "enzyme_prediction.txt" file was used for the performance evaluation. Finally, the 393 sequences that had between 50 and 5000 residues were uploaded to the webserver link provided in DEEPre's publication, http://www.cbrc.kaust.edu.sa/DEEPre/index.html. The prediction was considered non-enzymatic if the first digit was "0" and enzymatic for all other first digits.

## Data availability

The dataset and T-metal-site are available in the supplementary file Supplementary Data 2. T-metal-seq is also available in the supplemental file Supplementary Data 3. Other data are available from the corresponding author upon reasonable request.

## Code availability

The code used to optimize, select, and evaluate MAHOMES and the final saved models can be found at https://github.com/SluskyLab/MAHOMES[68].

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

## Acknowledgements

We gratefully acknowledge helpful conversations and paper edits from members of the Slusky lab as well as Yusuf Adeshina and Adam Pogorny. We appreciate Patrick Mahomes for inspiring the name of our most valuable predictor (MVP). NIH awards DP2GM128201, T32-GM008359, NSF award MCB160205, and The Gordon and Betty Moore Inventor Fellowship provided funding that supported this work.

## Author contributions

J.S.G.S., M.W.F., and R.F. conceived of the idea. R.F. developed the dataset; M.W.F. initiated the development of the algorithms; R.F. completed and benchmarked the algorithms. Writing and editing by J.S.G.S., M.W.F., and R.F.

## Competing interests

The authors declare no competing interests.
