## [Peer Review File · Nature Communications]

REVIEWER COMMENTS

Reviewer #1 (Remarks to the Author):

The authors have tested multiple machine learning methods to develop a high-performing classifier that determines whether a metal binding site in a protein structure is enzymatic or non-enzymatic. This represents a novel approach to an important and timely problem. The authors also delineate which input features are the most important for proper classification, offering insights into the microscopic features of enzymatic metal-binding sites. They also uncover some likely incorrect annotations. While the paper is generally clear, the only part that needs to be strengthened is the justification for their method of labeling of the set of proteins used for training and validation. Their method is based on homologues in the M-CSA database - the authors need to make a stronger argument that this approach is valid. Is functionality necessarily conserved across homologues? Whether the method is accepted in the community as a reliable predictor of enzymatic metal binding sites, or is viewed as simply predicting their labeling scheme, depends on the strength of this argument. The Materials and Methods section is sufficiently clear and detailed to allow for results to be reproduced, and the method utilized, by others.

Mary Jo Ondrechen

Reviewer #2 (Remarks to the Author):

In this paper, the authors present a model to differentiate enzymatic and non-enzymatic metals in proteins using a machine learning approach. The work is clearly presented and well described, and the results are interesting. However, I have a number of concerns regarding the construction and the use of the data sets. Specifically: (1) The workflow shown in Figure 1 seems to indicate that redundancy is removed after separation into the training set and the test set; if so, this may cause some sequences to be present in both sets, thereby affecting prediction results. The authors should clarify in more detail how the sets were built. (2) In association with point (1), the authors should also clarify the meaning of the Jaccard index, which they used to remove redundancy among sites; this is especially important, since it appears that they did not use sequence identity (except for 100% identical sequences) as a criterion to remove redundancy. (3) It seems that the authors used only one test set to evaluate the performance of the method; however, a good practice is to build several different test sets, and corresponding training sets, and average the results obtained in each case.

Provided that the authors address the above points, I consider the work suitable for publication in Nature Communications.

Reviewer #3 (Remarks to the Author):

The authors present a machine learning classifier capable of distinguishing metal binding sites involved in catalysis from metal binding sites not involved in catalysis. They also provide an interesting analysis of the specific properties of protein sites that best distinguish catalytic sites from non-catalytic sites.

The work is novel, and the generated models will be valuable to researchers in this field. The authors note that they intend to make the code required to generate the models available on GitHub upon publication - purely for the sake of making it as easy as possible to take

advantage of the research presented here, it might be even more beneficial to make the generated MAHOMES model itself available, either as a pickled Python object or by simply providing the trained model's parameters. This is not necessary to the publication of the paper itself, but might make it much easier for others to utilise the new model.

In the fifth paragraph of the introduction, the authors refer to imbalanced datasets and state 'such imbalanced datasets result in low precision'. While I understand and agree with the point I think the authors are making here, I think this sentence could be slightly reworded as it somewhat implies that precision is a metric which, like accuracy, is unsuitable in imbalanced datasets. In fact precision and recall are very useful in imbalanced datasets because, when used in tandem, they ensure that a classifier does not 'cheat' by always predicting the more numerous class. For datasets with more negative samples than positive samples such as the ones being referred to here, recall is indeed more important than precision, but the inverse is true for datasets with more positive samples than negatives. I think the sentence would benefit by being modified to stress that they are specifically referring to these kinds of imbalanced datasets, not imbalanced datasets generally. I appreciate the word 'such' does somewhat imply that they are referring only to this kind of dataset but, given the importance of this concept to the later Discussion, out of an abundance of caution I think it should be made even more clear.

The pipeline the authors built for building their test set does appear to be robust, and discards structures for which it has a reasonable degree of uncertainty over which class to allocate it to. However this pipeline is essentially itself a predictive model with its own precision and recall, and all the downstream analysis of and metrics for the author's final model are dependent on the ability of this initial classifier to correctly label structures. I would prefer to see a little more discussion of this key point, particularly as the authors did have to remove certain misclassified structures after manual inspection. There is no particular reason to suspect that the algorithm for creating the dataset is wildly inaccurate, but addressing this point early on the discussion would be more reassuring to the reader when reading the analysis done downstream of this crucial first step.

Finally, when discussing the algorithm for identifying sites and the criteria used for discarding sites, the authors do not mention any criteria based on the number of liganding atoms to the metal. This is a very important quality metric when assessing a metal binding site, as sites with few liganding atoms (one or perhaps two) may well not be a physiologically relevant, tightly bound binding site, but rather a loosely bound artefact of crystallisation. It may even be that the 'full' binding site is present only in the biological assembly and that the raw coordinates of the PDB contain only one binding residue of the actual site - though as the authors discard sites composed of multiple chains, this possibility is perhaps less important.

I found the paper to be well written and flowed logically, with the following very minor typos:

- The comma in the fifth sentence of the first paragraph in the Results section appears to be unnecessary ("...were placed in the dataset, used for algorithm optimization...").
- "usea" should be "uses" in the second paragraph of the 'Benchmarking other methods' subsection.

Response to reviewers:

We thank all three reviewers for their insightful and constructive comments and suggestions. We very much appreciate that your suggestions have made our manuscript stronger.

Reviewer #1 (Remarks to the Author):

1.1 While the paper is generally clear, the only part that needs to be strengthened is the justification for their method of labeling of the set of proteins used for training and validation. Their method is based on homologues in the M-CSA database - the authors need to make a stronger argument that this approach is valid. Is functionality necessarily conserved across homologues? Whether the method is accepted in the community as a reliable predictor of enzymatic metal binding sites, or is viewed as simply predicting their labeling scheme, depends on the strength of this argument.

Thank you for helping us strengthen this justification. In order to address this concern we have added three parts, 1) an estimate of how common M-CSA homologs are enzymes, 2) a manual inspection of the accuracy of our labeling, and 3) further manual inspection of sites labeled catalytic in our test set that did not have an associated EC number.

We have added a discussion and reference estimating how common homologues of M-CSA sites are not enzymes.

... It is estimated that between 0.5% and 4% of M-CSA homologues are likely to be pseudoenzymes, or non-catalytic enzyme homologs⁵². To avoid labeling pseudoenzymes, we discarded homologues that did not meet at least one of the following requirements; an associated EC number, a PDB classification containing “ase”, or a PDB Macromolecule Name containing “ase”.

In addition, in order to increase confidence in our labeling pipeline we added an estimation of our labeling accuracy with a manual inspection of 100 sites. The sites used and references checked for the manual inspection are now listed in Table ST1.

Our dataset used for ML is composed of 3,465 sites from 2,626 different PDBs; 24% of the sites are enzymatic (Supplementary file sites.csv). To check the accuracy of our pipeline labeling, we manually examined 50 sites labeled enzymatic and 50 sites labeled non-enzymatic. We found that all 50 sites labeled enzymatic were indeed at catalytic active sites. Three of the sites labeled non-enzymatic were really catalytic sites, giving our pipeline an estimated 97% balanced accuracy (Table ST1). The test-set, T-metal-site, which is mutually exclusive from the dataset is composed of 520 sites from 404 different PDBs; 31% of the sites are enzymatic. Both sets contain sites distributed among the six major EC classes (Figure S1) excluding the translocases a class added to the EC after the start of this project.

Finally, we had also manually inspected all enzyme-labeled sequences in T-metal-seq which did not have an associated EC number. We did not mention this in the initial draft, but included it now as additional support for the strength of our labeling scheme.

... sequences were labeled as enzymatic if they met at least one of the following criteria: an associated EC number, an “ase” suffix in structure classification or molecular name, or previously identified to be enzymatic by manual inspection. Sequences that did not meet any of the previous criteria were labeled as non-enzymatic. We manually verified 33 enzymatic labels for sequences that lacked an associated EC number. The sequence test-set contained 170 enzymatic sequences and 230 non-enzymatic sequences.

Reviewer #2 (Remarks to the Author):

2.(1) The workflow shown in Figure 1 seems to indicate that redundancy is removed after separation into the training set and the test set; if so, this may cause some sequences to be present in both sets, thereby affecting prediction results. The authors should clarify in more detail how the sets were built.

We agree that it is very important that no sequence/site is present in both sets. In the text accompanying figure 1, we elaborated on our previous wording to make sure that this is clear.

Biologically redundant data has shown to negatively impact machine learning models.³⁵ Having similar data in both the training and testing set can also lead to inflated performance evaluations. To prevent these issues, similarity within and between the test-set and dataset was reduced by filtering out sites with sequence or structural similarity. Briefly, within each set, sites were grouped according to homology. Within homologous groups the metal-binding sites are compared by similarity of coordinating residues. Only sites with unique residues were selected to represent each group. In this manner, homologs that have only mutated at the active site to perform different reactions will not be removed, but homologs with the same, conserved active site will be removed. Then in order to prevent similar proteins from being in both the data set and test set, we exhaustively compared all sites regardless of which set they were in, removing identical proteins and sites that are similar within a 6 Å sphere of the coordinated metal.

2.(2) In association with point (1), the authors should also clarify the meaning of the Jaccard index, which they used to remove redundancy among sites; this is especially important, since it appears that they did not use sequence identity (except for 100% identical sequences) as a criterion to remove redundancy.

To help the reader understand how Jaccard index is used to calculate structural similarity, we have added the following equation to the methods:

... We defined local site similarity as the Jaccard index of residue identities within a 3.75 Å radius for two sites. The Jaccard index is a similarity metric defined by Eq. 1, where **A** and **B** are the vector of amino acid identities surrounding two different sites.

$$J(\mathbf{A}, \mathbf{B}) = \frac{|\mathbf{A} \cap \mathbf{B}|}{|\mathbf{A} \cup \mathbf{B}|} \quad (1)$$

This results in a value of 0 when there is no similarity, and a value of 1 for identical sites. Sites were removed if they had a local similarity greater than 0.80. ...

In order to address how our redundancy removal ensures we do not have similar sites even when not explicitly checking for sequence similarity we have added the following language (as described in the point above as well).

Briefly, within each set, sites were grouped according to homology. Within homologous groups the metal-binding sites are compared by similarity of coordinating residues. Only sites with unique residues were selected to represent each group. In this manner, homologs that have only mutated at the active site to perform different reactions will not be removed, but homologs with the same, conserved active site will be removed.

2.(3) It seems that the authors used only one test set to evaluate the performance of the method; however, a good practice is to build several different test sets, and corresponding training sets, and average the results obtained in each case.

We thoroughly agree with the importance of training on several different training sets and test sets. Since our holdout test-set evaluation was the least likely to be overfit, it is the evaluation that we refer to throughout the paper. However, an evaluation like the one suggested here was also performed and was in agreement with our holdout test-set performance evaluation. We have revised the text to make this clearer.

Machine learning model optimization and selection

...

When a model learns the details of a training set too well it can negatively impact the model performance on new data. This is called overfitting. Cross-validation (CV) is common strategy which splits data into several different groups. Iteratively, one group at a time is left out for evaluation while the rest of the data is used for training. We used two CV techniques in a nested fashion (Figure S4), which allowed us to use different data for model optimization and model selection. Only the dataset was used for model optimization and model selection, allowing the test-set to act as a final model evaluation which was not be influenced by any previous training. During the inner CV, we optimized each algorithm for a specific feature set. We tested four different scoring metrics for optimization: accuracy, precision, MCC, or a multi-score combination of accuracy, MCC and Jaccard index. In total, 3,752 models were created (14 algorithms x 67 feature subsets x 4 optimization scoring metrics). We used the results from the outer CV to select the best of these models. However, 3,274 of the models used different “optimal” versions of the machine learning algorithm during the outer CV. To eliminate any inflated metrics that may have come from this, we re-ran the outer CV using only the most frequently selected version of the algorithm for each model and discarded all models where large deviations persisted (Methods and Figure S5).

...

Top model evaluation

... A final evaluation of MAHOMES was performed using the T-metal-site test-set (2018-2020 structures) where it achieved slightly higher performance metrics than its outer CV performance (Table ST4). The final performance evaluation still falls within projected deviation, as observed on different test folds during outer CV, supporting the validity of the reported performance metrics.

...

Machine learning

...

A nested cross-validation strategy was used for model optimization to avoid overfitting and bias. Each inner loop used GridSearch with StratifiedShuffleSplit (in the python scikit-learn package⁶⁷) and was optimized four times for each of four scoring terms – accuracy, precision, MCC, or a multi-score combination of accuracy, MCC and Jaccard index. The outer loop CV used a stratified k-fold cross validation. The most frequently selected hyperparameter set during the outer cross validation was considered optimal for the model. The dataset was under-sampled once prior to model optimization.

In total, we examined 3,752 machine learning models (14 algorithms x 67 feature sets x 4 optimization terms). For model selection, we re-ran the outer cross validation using only the optimal hyper-parameter set. During stratified k-fold cross validation, the data is divided into k groups (k = 7), each with the same number of positive and negative entries. All except for one of the groups are used to train a model and the left out group is used to evaluate that model. This is repeated k times, leaving out a different group each time, essentially testing the model on 7 different subsets. The performance is then averaged. Our random-sampling, and some of the machine learning algorithms that require random sampling, are susceptible to differences in the machines on which they are executed. In order to produce more reliable performance evaluations for model selection, we repeated each iteration of the outer cross validation ten times when we re-ran it. During each repetition, a new random seed was passed to the machine learning algorithm and used to under sample the training folds. Since we used k=7, the reported outer cross validation results are the average of 70 different models (7 folds, each with 10 different random seeds).

Reviewer #3 (Remarks to the Author):

3.1 The work is novel, and the generated models will be valuable to researchers in this field. The authors note that they intend to make the code required to generate the models available on GitHub upon publication - purely for the sake of making it as easy as possible to take advantage of the research presented here, it might be even more beneficial to make the generated MAHOMES model itself available, either as a pickled Python object or by simply providing the trained model's parameters. This is not necessary to the publication of the paper itself, but might make it much easier for others to utilise the new model.

Thank you for the suggestion. The generated models, saved as python pickles have been added to the GitHub (<https://github.com/SluskyLab/MAHOMES>).

3.2 *In the fifth paragraph of the introduction, the authors refer to imbalanced datasets and state 'such imbalanced datasets result in low precision'. While I understand and agree with the point I think the authors are making here, I think this sentence could be slightly reworded as it somewhat implies that precision is a metric which, like accuracy, is unsuitable in imbalanced datasets. In fact precision and recall are very useful in imbalanced datasets because, when used in tandem, they ensure that a classifier does not 'cheat' by always predicting the more*

numerous class. For datasets with more negative samples than positive samples such as the ones being referred to here, recall is indeed more important than precision, but the inverse is true for datasets with more positive samples than negatives. I think the sentence would benefit by being modified to stress that they are specifically referring to these kinds of imbalanced datasets, not imbalanced datasets generally. I appreciate the word 'such' does somewhat imply that they are referring only to this kind of dataset but, given the importance of this concept to the later Discussion, out of an abundance of caution I think it should be made even more clear.

We have reworded the sentence to explicitly state the type of imbalance being addressed:

... Such imbalanced datasets, where the positive samples are the minority class, result in low precision. ...

3.3 The pipeline the authors built for building their test set does appear to be robust, and discards structures for which it has a reasonable degree of uncertainty over which class to allocate it to. However this pipeline is essentially itself a predictive model with its own precision and recall, and all the downstream analysis of and metrics for the author's final model are dependent on the ability of this initial classifier to correctly label structures. I would prefer to see a little more discussion of this key point, particularly as the authors did have to remove certain misclassified structures after manual inspection. There is no particular reason to suspect that the algorithm for creating the dataset is wildly inaccurate, but addressing this point early on the discussion would be more reassuring to the reader when reading the analysis done downstream of this crucial first step.

We very much appreciate this point. To better assess the pipeline's predictive capability, we performed a manually inspection of 100 sites and determined that the pipeline had a balanced accuracy of 97%. The sites and literature that were inspected are enumerated in table ST1:

Our dataset used for ML is composed of 3,465 sites from 2,626 different PDBs; 24% of the sites are enzymatic (Supplementary file sites.csv). To check the accuracy of our pipeline labeling, we manually examined 50 sites labeled enzymatic and 50 sites labeled non-enzymatic. We found that all 50 sites labeled enzymatic were indeed at catalytic active sites. Three of the sites labeled non-enzymatic were really catalytic sites, giving our pipeline an estimated 97% balanced accuracy (Table ST1). The test-set, T-metal-site, which is mutually exclusive from the dataset is composed of 520 sites from 404 different PDBs; 31% of the sites are enzymatic. Both sets contain sites distributed among the six major EC classes (Figure S1) excluding the translocases a class added to the EC after the start of this project.

3.4 Finally, when discussing the algorithm for identifying sites and the criteria used for discarding sites, the authors do not mention any criteria based on the number of liganding atoms to the metal. This is a very important quality metric when assessing a metal binding site, as sites with few liganding atoms (one or perhaps two) may well not be a physiologically relevant, tightly bound binding site, but rather a loosely bound artefact of crystallisation. It may even be that the 'full' binding site is present only in the biological assembly and that the raw coordinates of the PDB contain only one binding residue of the actual site - though as the authors discard sites composed of multiple chains, this possibility is perhaps less important.

We agree that unliganded or few-liganded metals are likely crystal artifacts and are less useful for the non-enzymatic test set. We found that unliganded and few-liganded metals were likely to either dramatically move while preprocessing the structure with relaxation or fail our feature calculations. In either case, they are not used in the set. Ultimately, this results in the distribution of ligand atoms between enzyme and non-enzyme sets being remarkably similar. We have elaborated on our explanation in the methodology and added a supplementary figure to illustrate the result of these step.

Finally, sites in which the metal is not well coordinated are likely due to crystal artifacts and are poor negative controls for metalloenzyme sites. When the structures were relaxed using Rosetta (see Supplement for RosettaScripts inputs), 728 sites with loosely bound metals—often the result of crystal artifacts—that moved more than 3 Å during relaxation were removed from the dataset and test-set. Also, 179 sites were removed due to failure to complete feature calculations. The remaining metals had similar distribution of number of coordinating atoms between enzymatic and non-enzymatic sites (Figure S10).

Supplemental Figure S10: Coordinating atoms for enzyme and non-enzymes.

Comparison of the average number of N, O, and S liganding residue atoms per site for enzyme and non-enzyme sites.

I found the paper to be well written and flowed logically, with the following very minor typos:

3.5 - The comma in the fifth sentence of the first paragraph in the Results section appears to be unnecessary ("...were placed in the dataset, used for algorithm optimization...").

Fixed

3.6 - "usea" should be "uses" in the second paragraph of the 'Benchmarking other methods' subsection.

Fixed

REVIEWER COMMENTS

Reviewer #2 (Remarks to the Author):

The authors have answered all the doubts raised: the paper is good for publication

Reviewer #3 (Remarks to the Author):

The authors have addressed all my comments - I recommend the paper for publication.